# Practical Locally Private Heavy Hitters

Raef Bassily[*]    Kobbi Nissim[†]    Uri Stemmer[‡]    Abhradeep Thakurta[§]

## Abstract

We present new practical local differentially private heavy hitters algorithms achieving optimal or near-optimal worst-case error – TreeHist and Bitstogram. In both algorithms, server running time is $\tilde{O}(n)$ and user running time is $\tilde{O}(1)$, hence improving on the prior state-of-the-art result of Bassily and Smith [STOC 2015] requiring $\tilde{O}(n^{5/2})$ server time and $\tilde{O}(n^{3/2})$ user time. With a typically large number of participants in local algorithms ($n$ in the millions), this reduction in time complexity, in particular at the user side, is crucial for the use of such algorithms in practice. We implemented Algorithm TreeHist to verify our theoretical analysis and compared its performance with the performance of Google's RAPPOR code.

## 1 Introduction

We revisit the problem of computing heavy hitters with local differential privacy. Such computations have already been implemented to provide organizations with valuable information about their user base while providing users with the strong guarantee that their privacy would be preserved even if the organization is subpoenaed for the entire information seen during an execution. Two prominent examples are Google's use of RAPPOR in the Chrome browser [10] and Apple's use of differential privacy in iOS-10 [16]. These tools are used for learning new words typed by users and identifying frequently used emojis and frequently accessed websites.

**Differential privacy in the local model.** Differential privacy [9] provides a framework for rigorously analyzing privacy risk and hence can help organization mitigate users' privacy concerns as it ensures that what is learned about any individual user would be (almost) the same whether the user's information is used as input to an analysis or not.

Differentially private algorithms work in two main modalities – the *curator model* and the *local model*. The curator model assumes a trusted centralized curator that collects all the personal information and then analyzes it. The local model on the other hand, does not involve a central repository. Instead, each piece of personal information is randomized by its provider to protect privacy even if all information provided to the analysis is revealed. Holding a central repository of personal information can become a liability to organizations in face of security breaches, employee misconduct, subpoenas, etc. This makes the local model attractive for implementation. Indeed in the last few years Google and Apple have deployed local differentially private analyses [10, 16].

**Challenges of the local model.** A disadvantage of the local model is that it requires introducing noise at a significantly higher level than what is required in the curator model. Furthermore, some tasks which are possible in the curator model are impossible in the local model [9, 14, 7]. To see the effect of noise, consider estimating the number of HIV positives in a given population of $n$ participants. In the curated model, it suffices to add Laplace noise of magnitude $O(1/\epsilon)$ [9], i.e.,

---

[*]Department of Computer Science & Engineering, The Ohio State University. `bassily.1@osu.edu`

[†]Department of Computer Science, Georgetown University. `kobbi.nissim@georgetown.edu`

[‡]Center for Research on Computation and Society (CRCS), Harvard University. `u@uri.co.il`

[§]Department of Computer Science, University of California Santa Cruz. `aguhatha@ucsc.edu`.

independent of $n$. In contrast, a lowerbound of $\Omega(\sqrt{n}/\epsilon)$ is known for the local model [7]. A higher noise level implies that the number of participants $n$ needs to be large (maybe in the millions for a reasonable choice of $\epsilon$). An important consequence is that practical local algorithms must exhibit low time, space, and communication complexity, especially at the user side. This is the problem addressed in our work.

**Heavy hitters and histograms in the local model.** Assume each of $n$ users holds an element $x_i$ taken from a domain of size $d$. A histogram of this data lists (an estimate of) the multiplicity of each domain element in the data. When $d$ is large, a succinct representation of the histogram is desired either in form of a *frequency oracle* – allowing to approximate the multiplicity of any domain element – and *heavy hitters* – listing the multiplicities of most frequent domain elements, implicitly considering the multiplicities of other domain elements as zero. The problem of computing histograms with differential privacy has attracted significant attention both in the curator model [9, 5, 6] and the local model [13, 10, 4]. Of relevance is the work in [15].

We briefly report on the state of the art heavy hitters algorithms of Bassily and Smith [4] and Thakurta et al. [16], which are most relevant for the current work. Bassily and Smith provide matching lower and upper bounds of $\Theta(\sqrt{n \log(d)}/\epsilon)$ on the worst-case error of local heavy hitters algorithms. Their local algorithm exhibits optimal communication but a rather high time complexity: Server running time is $\tilde{O}(n^{5/2})$ and, crucially, user running time is $\tilde{O}(n^{3/2})$ – complexity that severely hampers the practicality of this algorithm. The construction by Thakurta et al. is a heuristic with no bounds on server running time and accuracy.[1] User computation time is $\tilde{O}(1)$, a significant improvement over [4]. See Table 1.

**Our contributions.** The focus of this work is on the design of locally private heavy hitters algorithms with near optimal error, keeping time, space, and communication complexity minimal. We provide two new constructions of heavy hitters algorithms TreeHist and Bitstogram that apply different techniques and achieve similar performance. We implemented Algorithm TreeHist and provide measurements in comparison with RAPPOR [10] (the only currently available implementation for local histograms). Our measurements are performed with a setting that is favorable to RAPPOR (i.e., a small input domain), yet they indicate that Algorithm TreeHist performs better than RAPPOR in terms of noise level.

Table 1 details various performance parameters of algorithms TreeHist and Bitstogram, and the reader can check that these are similar up to small factors which we ignore in the following discussion. Comparing with [4], we improve time complexity both at the server (reduced from $\tilde{O}(n^{5/2})$ to $\tilde{O}(n)$) and at the user (reduced from $\tilde{O}(n^{3/2})$ to $O(\max{(\log n, \log d)}^2)$). Comparing with [16], we get provable bounds on the server running time and worst-case error. Note that Algorithm Bitstogram achieves optimal worst-case error whereas Algorithm TreeHist is almost optimal, by a factor of $\sqrt{\log(n)}$.

| Performance metric | TreeHist (this work) | Bitstogram (this work) | Bassily and Smith [4][2] |
|---|---|---|---|
| Server time | $\tilde{O}(n)$ | $\tilde{O}(n)$ | $\tilde{O}(n^{5/2})$ |
| User time | $\tilde{O}(1)$ | $\tilde{O}(1)$ | $\tilde{O}(n^{3/2})$ |
| Server processing memory | $\tilde{O}(\sqrt{n})$ | $\tilde{O}(\sqrt{n})$ | $O(n^2)$ |
| User memory | $\tilde{O}(1)$ | $\tilde{O}(1)$ | $\tilde{O}(n^{3/2})$ |
| Communication/user | $O(1)$ | $O(1)$ | $O(1)$ |
| Public randomness/user [3] | $\tilde{O}(1)$ | $\tilde{O}(1)$ | $\tilde{O}(n^{3/2})$ |
| Worst-case Error | $O\left(\sqrt{n \log(n) \log(d)}\right)$ | $O\left(\sqrt{n \log(d)}\right)$ | $O\left(\sqrt{n \log(d)}\right)$ |

Table 1: Achievable performance of our protocols, and comparison to the prior state-of-the-art by Bassily and Smith [4]. For simplicity, the $\tilde{O}$ notation hides logarithmic factors in $n$ and $d$. Dependencies on the failure probability $\beta$ and the privacy parameter $\epsilon$ are omitted.

**Elements of the constructions.** Main details of our constructions are presented in sections 3 and 4. Both our algorithms make use of frequency oracles – data structures that allow estimating various counts.

Algorithm TreeHist identifies heavy-hitters and estimates their frequencies by scanning the levels of a binary prefix tree whose leaves correspond to dictionary items. The recovery of the heavy hitters is in a bit-by-bit manner. As the algorithm progresses down the tree it prunes all the nodes that *cannot* be prefixes of heavy hitters, hence leaving $\tilde{O}(\sqrt{n})$ nodes in every depth. This is done by making queries to a frequency oracle. Once the algorithm reaches the final level of the tree it identifies the list of heavy hitters. It then invokes the frequency oracle once more on those particular items to obtain more accurate estimates for their frequencies.

Algorithm Bitstogram hashes the input domain into a domain of size roughly $\sqrt{n}$. The observation behind this algorithm is that if a heavy hitter $x$ does not collide with other heavy hitters then $(h(x), x_i)$ would have a significantly higher count than $(h(x), \neg x_i)$ where $x_i$ is the $i$th bit of $x$. This allows recovering all bits of $x$ in parallel given an appropriate frequency oracle.

We remark that even though we describe our protocols as operating in phases (e.g., scanning the levels of a binary tree), these phases are done in parallel, and our constructions are *non-interactive*. All users participate simultaneously, each sending a single message to the server. We also remark that while our focus is on algorithms achieving the optimal (i.e., smallest possible) error, our algorithms are also applicable when the server is interested in a larger error, in which case the server can choose a random subsample of the users to participate in the computation. This will reduce the server runtime and memory usage, and also reduce the privacy cost in the sense that the unsampled users get perfect privacy (so the server might use their data in another analysis).

## 2 Preliminaries

### 2.1 Definitions and Notation

**Dictionary and users items:** Let $\mathcal{V} = [d]$. We consider a set of $n$ users, where each user $i \in [n]$ has some item $v_i \in \mathcal{V}$. Sometimes, we will also use $v_i$ to refer to the binary representation $v_i$ when it is clear from the context.

**Frequencies:** For each item $v \in \mathcal{V}$, we define the frequency $f(v)$ of such item as the number of users holding that item, namely, $f(v) \triangleq \sum_{i \in [n]} \mathbf{1}(v_i = v)$, where $\mathbf{1}(E)$ of an event $E$ is the indicator function of $E$.

**A frequency oracle:** is a data structure together with an algorithm that, for any given $v \in \mathcal{V}$, allows computing an estimate $\hat{f}(v)$ of the frequency $f(v)$.

**A succinct histogram:** is a data structure that provides a (short) list of items $\hat{v}_1, ..., \hat{v}_k$, called the *heavy hitters*, together with estimates for their frequencies $(\hat{f}(\hat{v}_j) : j \in [k])$. The frequencies of the items not in the list are implicitly estimated as $\hat{f}(v) = 0$. We measure the error in a succinct histogram by the $\ell_\infty$ distance between the estimated and true frequencies, $\max_{v \in [d]} |\hat{f}(v) - f(v)|$. We will also consider the maximum error in the estimated frequencies restricted to the items in the list, that is, $\max_{\hat{v}_j : j \in [k]} |\hat{f}(\hat{v}_j) - f(\hat{v}_j)|$.

If a data succinct histogram aims to provide $\ell_\infty$ error $\eta$, the list does not need to contain more than $O(1/\eta)$ items (since items with estimated frequencies below $\eta$ may be omitted from the list, at the price of at most doubling the error).

## 2.2 Local Differential Privacy

In the local model, an algorithm $\mathcal{A} : \mathcal{V} \to \mathcal{Z}$ accesses the database $\mathbf{v} = (v_1, \cdots, v_n) \in \mathcal{V}^n$ only via an oracle that, given any index $i \in [n]$, runs a local randomized algorithm (local randomizer) $\mathcal{R} : \mathcal{V} \to \tilde{\mathcal{Z}}$ on input $v_i$ and returns the output $\mathcal{R}(v_i)$ to $\mathcal{A}$.

**Definition 2.1** (Local differential privacy [9, 11]). *An algorithm satisfies $\epsilon$-local differential privacy (LDP) if it accesses the database $\mathbf{v} = (v_1, \cdots, v_n) \in \mathcal{V}^n$ only via invocations of a local randomizer $\mathcal{R}$ and if for all $i \in [n]$, if $\mathcal{R}^{(1)}, \ldots, \mathcal{R}^{(k)}$ denote the algorithm's invocations of $\mathcal{R}$ on the data sample $v_i$, then the algorithm $\mathcal{A}(\cdot) \triangleq \left( \mathcal{R}^{(1)}(\cdot), \mathcal{R}^{(2)}(\cdot), \ldots, \mathcal{R}^{(k)}(\cdot) \right)$ is $\epsilon$-differentially private. That is, if for any pair of data samples $v, v' \in \mathcal{V}$ and $\forall \mathcal{S} \subseteq \mathsf{Range}(\mathcal{A})$, $\Pr[\mathcal{A}(v) \in \mathcal{S}] \le e^{\epsilon} \Pr[\mathcal{A}(v') \in \mathcal{S}]$.*

## 3 The TreeHist Protocol

In this section, we briefly give an overview of our construction that is based on a compressed, noisy version of the count sketch. To maintain clarity of the main ideas, we give here a high-level description of our construction. We refer to the full version of this work [3] for a detailed description of the full construction.

We first introduce some objects and public parameters that will be used in the construction:

**Prefixes:** For a binary string $v$, we will use $v[1 : \ell]$ to denote the $\ell$-bit prefix of $v$. Let $\overline{\mathcal{V}} = \{ v \in \{0, 1\}^{\ell} \text{ for some } \ell \in [\log d] \}$. Note that elements of $\overline{\mathcal{V}}$ arranged in a binary prefix tree of depth $\log d$, where the nodes at level $\ell$ of the tree represent all binary strings of length $\ell$. The items of the dictionary $\mathcal{V}$ represent the bottommost level of that tree.

**Hashes:** Let $t, m$ be positive integers to be specified later. We will consider a set of $t$ pairs of hash functions $\{(h_1, g_1), \ldots, (h_t, g_t)\}$, where for each $i \in [t]$, $h_i : \overline{\mathcal{V}} \to [m]$ and $g_i : \overline{\mathcal{V}} \to \{-1, +1\}$ are independently and uniformly chosen pairwise independent hash functions.

**Basis matrix:** Let $\mathbf{W} \in \{-1, +1\}^{m \times m}$ be $\sqrt{m} \cdot \mathbf{H}_m$ where $\mathbf{H}_m$ is the Hadamard transform matrix of size $m$. It is important to note that we do not need to store this matrix. The value of any entry in this matrix can be computed in $O(\log m)$ bit operations given the (row, column) index of that entry.

**Global parameters:** The total number of users $n$, the size of the Hadamard matrix $m$, the number of hash pairs $t$, the privacy parameter $\epsilon$, the confidence parameter $\beta$, and the hash functions $\{(h_1, g_1), \ldots, (h_t, g_t)\}$ are assumed to be public information. We set $t = O(\log(n/\beta))$ and $m = O\left( \sqrt{\frac{n}{\log(n/\beta)}} \right)$.

**Public randomness:** In addition to the $t$ hash pairs $\{(h_1, g_1), \ldots, (h_t, g_t)\}$, we assume that the server creates a random partition $\Pi : [n] \to [\log d] \times [t]$ that assigns to each user $i \in [n]$ a random pair $(\ell_i, j_i) \leftarrow [\log(d)] \times [t]$, and another random function $\mathcal{Q} : [n] \leftarrow [m]$ that assigns[4] to each user $i$ a uniformly random index $r_i \leftarrow [m]$. We assume that such random indices $\ell_i, j_i, r_i$ are shared between the server and each user.

First, we describe the two main modules of our protocol.

### 3.1 A local randomizer: LocalRnd

For each $i \in [n]$, user $i$ runs her own independent copy of a local randomizer, denoted as LocalRnd, to generate her private report. LocalRnd of user $i$ starts by acquiring the index triple $(\ell_i, j_i, r_i) \leftarrow [\log d] \times [t] \times [m]$ from public randomness. For each user, LocalRnd is invoked twice in the full protocol: once during the first phase of the protocol (called the pruning phase) where the high-frequency items (*heavy hitters*) are identified, and a second time during the final phase (the estimation phase) to enable the protocol to get better estimates for the frequencies of the heavy hitters.

In the first invocation, LocalRnd of user $i$ performs its computation on the $\ell_i$-th prefix of the item $v_i$ of user $i$, while in the second invocation, it performs the computation on the entire user's string $v_i$.

Apart from this, in both invocations, LocalRnd follows similar steps. It first selects the hash pair $(h_{j_i}, g_{j_i})$, computes $c_i = h_{j_i}(v_i[1 : \tilde{\ell}])$ (where $\tilde{\ell} = \ell_i$ in the first invocation and $\tilde{\ell} = \log d$ in the second invocation, and $v_i[1 : \tilde{\ell}]$ is the $\tilde{\ell}$-th prefix of $v_i$), then it computes a bit $x_i = g_{j_i}\left(v_i[1 : \tilde{\ell}]\right) \cdot W_{r_i, c_i}$ (where $W_{r,c}$ denotes the $(r, c)$ entry of the basis matrix $\mathbf{W}$). Finally, to guarantee $\epsilon$-local differential privacy, it generates a randomized response $y_i$ based on $x_i$ (i.e., $y_i = x_i$ with probability $e^{\epsilon/2}/(1 + e^{\epsilon/2})$ and $y_i = -x_i$ with probability $1/(1 + e^{\epsilon/2})$), which is sent to the server.

Our local randomizer can thought of as a transformed, compressed (via sampling), and randomized version of the count sketch [8]. In particular, we can think of LocalRnd as follows. It starts off with similar steps to the standard count sketch algorithm, but then deviates from it as it applies Hadamard transform to the user's signal, then samples one bit from the result. By doing so, we can achieve significant savings in space and communication without sacrificing accuracy.

### 3.2 A frequency oracle: FreqOracle

Suppose we want to allow the server estimate the frequencies of some *given* subset $\widehat{\mathcal{V}} \subseteq \{0, 1\}^\ell$ for some given $\ell \in [\log d]$ based on the noisy users' reports. We give a protocol, denoted as FreqOracle, for accomplishing this task.

For each queried item $\hat{v} \in \widehat{\mathcal{V}}$ and for each hash index $j \in [t]$, FreqOracle computes $c = h_j(\hat{v})$, then collects the noisy reports of a collection of users $\mathcal{I}_{\ell,j}$ that contains every user $i$ whose pair of prefix and hash indices $(\ell_i, j_i)$ match $(\ell, j)$. Next, it estimates the inverse Hadamard transform of the compressed and noisy signal of each user in $\mathcal{I}_{\ell,j}$. In particular, for each $i \in \mathcal{I}_{\ell,j}$, it computes $y_i W_{r_i, c}$ which can be described as a multiplication between $y_i \mathbf{e}_{r_i}$ (where $\mathbf{e}_{r_i}$ is the indicator vector with 1 at the $r_i$-th position) and the scaled Hadamard matrix $\mathbf{W}$, followed by selecting the $c$-th entry of the resulting vector. This brings us back to the standard count sketch representation. It then sums all the results and multiplies the outcome by $g_j(\hat{v})$ to obtain an estimate $\hat{f}_j(\hat{v})$ for the frequency of $\hat{v}$. As in the count sketch algorithm, this is done for every $j \in [t]$, then FreqOracle obtains a high-confidence estimate by computing the median of all the $t$ frequency estimates.

### 3.3 The protocol: TreeHist

The protocol is easier to describe via operations over nodes of the prefix tree $\overline{\mathcal{V}}$ of depth $\log d$ (described earlier). The protocol runs through two main phases: the pruning (or, scanning) phase, and the final estimation phase.

In the pruning phase, the protocol scans the levels of the prefix tree starting from the top level (that contains just 0 and 1) to the bottom level (that contains all items of the dictionary). For a given node at level $\ell \in [\log d]$, using FreqOracle as a subroutine, the protocol gets an estimate for the frequency of the corresponding $\ell$-bit prefix. For any $\ell \in [\log(d) - 1]$, before the protocol moves to level $\ell + 1$ of the tree, it prunes all the nodes in level $\ell$ that *cannot* be prefixes of actual heavy hitters (high-frequency items in the dictionary).Then, as it moves to level $\ell + 1$, the protocol considers only the children of the surviving nodes in level $\ell$. The construction guarantees that, with high probability, the number of surviving nodes in each level cannot exceed $O\left(\sqrt{\frac{n}{\log(d)\log(n)}}\right)$. Hence, the total number of nodes queried by the protocol (i.e., submitted to FreqOracle) is at most $O\left(\sqrt{\frac{n\log(d)}{\log(n)}}\right)$.

In the second and final phase, after reaching the final level of the tree, the protocol would have already identified a list of the candidate heavy hitters, however, their estimated frequencies may not be as accurate as we desire due to the large variance caused by the random partitioning of users across all the levels of the tree. Hence, it invokes the frequency oracle once more on those particular items, and this time, the sampling variance is reduced as the set of users is partitioned only across the $t$ hash pairs (rather than across $\log(d) \times t$ bins as in the pruning phase). By doing this, the server obtains more accurate estimates for the frequencies of the identified heavy hitters. The privacy and accuracy guarantees are stated below. The full details are given in the full version [3].

### 3.4 Privacy and Utility Guartantees

**Theorem 3.1.** Protocol TreeHist is $\epsilon$-local differentially private.

**Theorem 3.2.** There is a number $\eta = O\left(\sqrt{n \log(n/\beta) \log(d)}/\epsilon\right)$ such that with probability at least $1 - \beta$, the output list of the TreeHist protocol satisfies the following properties:

1. it contains all items $v \in \mathcal{V}$ whose true frequencies above $3\eta$.
2. it does not contain any item $v \in \mathcal{V}$ whose true frequency below $\eta$.
3. Every frequency estimate in the output list is accurate up to an error $\leq O\left(\sqrt{n \log(n/\beta)}/\epsilon\right)$

## 4 Locally Private Heavy-hitters – bit by bit

We now present a simplified description of our second protocol, that captures most of the ideas. We refer the reader to the full version of this work for the complete details.

**First Step: Frequency Oracle.** Recall that a frequency oracle is a protocol that, after communicating with the users, outputs a *data structure* capable of approximating the frequency of every domain element $v \in \mathcal{V}$. So, if we were to allow the server to have linear runtime in the domain size $|\mathcal{V}| = d$, then a frequency oracle would suffice for computing histograms. As we are interested in protocols with a significantly lower runtime, we will only use a frequency oracle as a subroutine, and query it only for (roughly) $\sqrt{n}$ elements.

Let $Z \in \{\pm 1\}^{d \times n}$ be a matrix chosen uniformly at random, and assume that $Z$ is publicly known.[5] That is, for every domain element $v \in \mathcal{V}$ and every user $j \in [n]$, we have a random bit $Z[v, j] \in \{\pm 1\}$. As $Z$ is publicly known, every user $j$ can identify its corresponding bit $Z[v_j, j]$, where $v_j \in \mathcal{V}$ is the input of user $j$. Now consider a protocol in which users send randomized responses of their corresponding bits. That is, user $j$ sends $y_j = Z[v_j, j]$ w.p. $\frac{1}{2} + \frac{\epsilon}{2}$ and sends $y_j = -Z[v_j, j]$ w.p. $\frac{1}{2} - \frac{\epsilon}{2}$. We can now estimate the frequency of every domain element $v \in \mathcal{V}$ as

$$a(v) = \frac{1}{\epsilon} \cdot \sum_{j \in [n]} y_j \cdot Z[v, j].$$

To see that $a(v)$ is accurate, observe that $a(v)$ is the sum of $n$ independent random variables (one for every user). For the users $j$ holding the input $v$ being estimated (that is, $v_j = v$) we will have that $\frac{1}{\epsilon}\mathbb{E}[y_j \cdot Z[v, j]] = 1$. For the other users we will have that $y_j$ and $Z[v, j]$ are independent, and hence $\mathbb{E}[y_j \cdot Z[v, j]] = \mathbb{E}[y_j] \cdot \mathbb{E}[Z[v, j]] = 0$. That is, $a(v)$ can be expressed as the sum of $n$ independent random variables: $f(v)$ variables with expectation 1, and $(n - f(v))$ variables with expectation 0. The fact that $a(v)$ is an accurate estimation for $f(v)$ now follows from the Hoeffding bound.

**Lemma 4.1** (Algorithm `Hashtogram`)**.** *Let $\epsilon \leq 1$. Algorithm* `Hashtogram` *satisfies $\epsilon$-LDP. Furthermore, with probability at least $1 - \beta$, algorithm* `Hashtogram` *answers every query $v \in \mathcal{V}$ with $a(v)$ satisfying:* $|a(v) - f(v)| \leq O\left(\frac{1}{\epsilon} \cdot \sqrt{n \log\left(\frac{nd}{\beta}\right)}\right).$

**Second Step: Identifying Heavy-Hitters.** Let us assume that we have a frequency oracle protocol with worst-case error $\tau$. We now want to use our frequency oracle in order to construct a protocol that operates on two steps: First, it identifies a small set of potential "heavy-hitters", i.e., domain elements that appear in the database at least $2\tau$ times. Afterwards, it uses the frequency oracle to estimate the frequencies of those potential heavy elements.[6]

Let $h : \mathcal{V} \to [T]$ be a (publicly known) random hash function, mapping domain elements into $[T]$, where $T$ will be set later.[7] We will now use $h$ in order to identify the heavy-hitters. To that end,

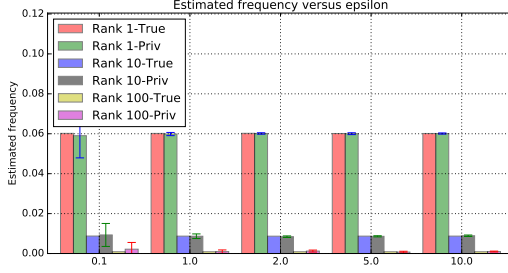

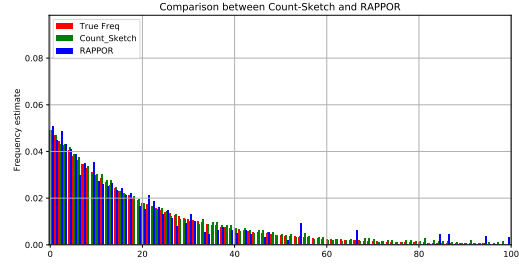

Figure 1: Frequency vs privacy ($\epsilon$) on the NLTK-Brown corpus.

Figure 2: Frequency vs privacy ($\epsilon$) on the Demo 3 experiment from RAPPOR

let $v^* \in \mathcal{V}$ denote such a heavy-hitter, appearing at least $2\tau$ times in the database $S$, and denote $t^* = h(v^*)$. Assuming that $T$ is big enough, w.h.p. we will have that $v^*$ is the only input element (from $S$) that is mapped (by $h$) into the hash value $t^*$. Assuming that this is indeed the case, we will now identify $v^*$ bit by bit.

For $\ell \in [\log d]$, denote $S_\ell = (h(v_j), v_j[\ell])_{j \in [n]}$, where $v_j[\ell]$ is bit $\ell$ of $v_j$. That is, $S_\ell$ is a database over the domain $([T] \times \{0, 1\})$, where the row corresponding to user $j$ is $(h(v_j), v_j[\ell])$. Observe that every user can compute her own row locally. As $v^*$ is a heavy-hitter, for every $\ell \in [\log d]$ we have that $(t^*, v^*[\ell])$ appears in $S_\ell$ at least $2\tau$ times. On the other hand, as we assumed that $v^*$ is the only input element that is mapped into $t^*$ we get that $(t^*, 1 - v^*[\ell])$ does not appear in $S_\ell$ at all. Recall that our frequency oracle has error at most $\tau$, and hence, we can use it to accurately determine the bits of $v^*$.

To make things more concrete, consider the protocol that for every hash value $t \in [T]$, for every coordinate $\ell \in [\log d]$, and for every bit $b \in \{0, 1\}$, obtains an estimation (using the frequency oracle) for the multiplicity of $(t, b)$ in $S_\ell$ (so there are $\log d$ invocations of the frequency oracle, and a total of $2T \log d$ estimations). Now, for every $t \in [T]$ let us define $\hat{v}^{(t)}$ where bit $\ell$ of $\hat{v}^{(t)}$ is the bit $b$ s.t. $(t, b)$ is more frequent than $(t, 1-b)$ in $S_\ell$. By the above discussion, we will have that $\hat{v}^{(t^*)} = v^*$. That is, the protocol identifies a set of $T$ domain elements, containing all of the heavy-hitters. The frequency of the identified heavy-hitters can then be estimated using the frequency oracle.

**Remark 4.1.** As should be clear from the above discussion, it suffices to take $T \gtrsim n^2$, as this will ensure that there are no collisions among different input elements. As we only care about collisions between "heavy-hitters" (appearing in $S$ at least $\sqrt{n}$ times), it would suffice to take $T \gtrsim n$ to ensure that w.h.p. there are no collisions between heavy-hitters. In fact, we could even take $T \gtrsim \sqrt{n}$, which would ensure that a heavy-hitter $x^*$ has no collisions with constant probability, and then to amplify our confidence using repetitions.

**Lemma 4.2** (Algorithm `Bitstogram`). *Let $\epsilon \leq 1$. Algorithm `Bitstogram` satisfies $\epsilon$-LDP. Furthermore, the algorithm returns a list $L$ of length $\tilde{O}(\sqrt{n})$ satisfying:*

1. *With probability $1 - \beta$, for every $(v, a) \in L$ we have that $|a - f(v)| \leq O\left(\frac{1}{\epsilon}\sqrt{n \log(n/\beta)}\right)$.*

2. *W.p. $1 - \beta$, for every $v \in \mathcal{V}$ s.t. $f(v) \geq O\left(\frac{1}{\epsilon}\sqrt{n \log(d/\beta) \log(\frac{1}{\beta})}\right)$, we have that $v$ is in $L$.*

## 5 Empirical Evaluation

We now discuss implementation details of our algorithms mentioned in Section 3[8]. The main objective of this section is to emphasize the empirical efficacy of our algorithms. [16] recently claimed space optimality for a similar problem, but a formal analysis (or empirical evidence) was not provided.

## 5.1 Evaluation of the Private Frequency Oracle

The objective of this experiment is to test the efficacy of our algorithm in estimating the frequencies of a known set of dictionary of user items, under local differential privacy. We estimate the error in estimation while varying the privacy parameter $\epsilon$. (See Section 2.1 for a refresher on the notation.) We ran the experiment (Figure 1) on a data set drawn uniformly at random from the NLTK Brown corpus [1]. The data set we created has $n = 10$ million samples drawn i.i.d. from the corpus with replacement (which corresponds to $25,991$ unique words), and the system parameters are chosen as follows: *number of data samples* $(n)$ : 10 million, range of the hash function $(m)$: $\sqrt{n}$, number of hash functions $(t)$: $285$. For the hash functions, we used the prefix bits of SHA-256. The estimated frequency is scaled by the number of samples to normalize the result, and each experiment is averaged over *ten runs*. In this plot, the rank corresponds to the rank of a domain element in the distribution of *true* frequencies in the data set. *Observations:* i) The plots corroborate the fact that the frequency oracle is indeed *unbiased*. The average frequency estimate (over ten runs) for each percentile is within one standard deviation of the corresponding true estimate. ii) The error in the estimates go down significantly as the privacy parameter $\epsilon$ is increased.

**Comparison with RAPPOR [10].** Here we compare our implementation with the only publicly available code for locally private frequency estimation. We took the snapshot of the RAPPOR code base (`https://github.com/google/rappor`) on May 9th, 2017. To perform a fair comparison, we tested our algorithm against one of the demo experiments available for RAPPOR (*Demo3* using the `demo.sh` script) with the same privacy parameter $\epsilon = \ln(3)$, the number of data samples $n = 1$ million, and the data set to be the same data set generated by the `demo.sh` script. In Figure 2 we observe that for higher frequencies both RAPPOR and our algorithm perform similarly, with ours being slightly better. However, in lower frequency regimes, the RAPPOR estimates are zero most of the times, while our estimates are closer to the true estimates. We do not claim our algorithm to be universally better than RAPPOR on all data sets. Rather, through our experiments we want to motivate the need for more thorough empirical comparison of both the algorihtms.

## 5.2 Private Heavy-hitters

In this section, we take on the harder task of identifying the heavy hitters, rather than estimating the frequencies of domain elements. We run our experiments on the NLTK data set described earlier, with the same default system parameters (as Section 5.1) along with $n = 10$ mi and $\epsilon = 2$, except now we assume that we do not know the domain. As a part of our algorithm design, we assume that every element in the domain is from the english alphabet set [a-z] and are of length exactly equal to *six* letters. Words longer than six letters were truncated and words shorter than six letters were tagged $\perp$ at the end. We set a threshold of $15 \cdot \sqrt{n}$ as the threshold for being a heavy hitter. As with moth natural language data sets, the NLTK Brown data follows a power law dirstribution with a very long tail. (See the full version of this work for a visualization of the distribution.)

In Table 5.2 we state our corresponding precision and recall parameters, and the false positive rate. The total number of positive examples is 22 (out of 25991 unique words),and the total number of negative examples is roughly $3 \times 10^8$. The total number of false positives $\mathsf{FP} = 60$, and false negatives $\mathsf{FN} = 3$. This corresponds to a vanishing $\mathsf{FP}$-rate, considering the total number of negative examples roughly equals $3 \times 10^8$. In practice, if there are false positives, they can be easily pruned using domain expertise. For example, if we are trying to identify new words which users are typing in English [2], then using the domain expertise of English, a set of false positives can be easily ruled out by inspecting the list of heavy hitters output by the algorithm. On the other hand, this cannot be done for false negatives. Hence, it is important to have a high recall value. The fact that we have three false negatives is because the frequency of those words are very close to the threshold of $15\sqrt{n}$. While there are other algorithms for finding heavy-hitters [4, 13], either they do not provide any theoretical guarantee for the utility [10, 12, 16], or there does not exist a scalable and efficient implementation for them.

| Data set | unique words | Precision | Recall (TPR) | FPR |
|---|---|---|---|---|
| NLTK Brown corpus | 25991 | 0.24 ($\sigma = 0.04$) | 0.86 ($\sigma = 0.05$) | $2 \times 10^{-7}$ |

Table 2: Private Heavy-hitters with threshold=$15\sqrt{n}$. Here $\sigma$ corresponds to the standard deviation. TPR and FPR correspond to true positive rate and false positive rates respectively.

## Footnotes

[1]The underlying construction in [16] is of a frequency oracle.

[2] The *user's* run-time and memory in [4] can be improved to $O(n)$ if one assumes random access to the public randomness, which we do not assume in this work.

[3] Our protocols can be implemented without public randomness while attaining essentially the same performance.

[4]We could have grouped $\Pi$ and $\mathcal{Q}$ into one random function mapping $[n]$ to $[\log d] \times [t] \times [m]$, however, we prefer to split them for clarity of exposition as each source of randomness will be used for a different role.

[5]As we describe in the full version of this work, $Z$ has a short description, as it need not be uniform.

[6]Event though we describe the protocol as having two steps, the necessary communication for these steps can be done in parallel, and hence, our protocol will have only 1 round of communication.

[7]As with the matrix $Z$, the hash function $h$ can have a short description length.

[8]The experiments are performed without the Hadamard compression during data transmission.

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
