[Reviews · NeurIPS 2017]

Reviewer 1



Overall, the improvement compared to the state-of-the-art [4] is not clearly stated. In Table 1, it would be better to show the comparison with existing methods. Does this work enable to reduce the server time and user time with achieving the same utility and privacy achieved by [4]? In Sec 3.2, what is the difference between FreqOracle protocol and the basic randomizer protocol in [4]? Line 187: In the TreeHist protocol, the high probability bound on the number of nodes survived is shown whereas the pruning strategy is not described at all. In algorithm 3 and algorithm 6, I found the pruning logic, but this point should be mentioned in the main body. I could not find justification on the high probability bound on the number of nodes in the appendix. In Section 5.1, the authors explain the results as follows: ...for higher frequencies both RAPPOR and our algorithm perform similarly. However, in lower frequency regimes, the RAPPOR estimates are zero most of the times, while our estimates are closer to the true estimates. This should be numerically evaluated. What is important in the heavy hitter problem is that the frequencies of higher rank elements are estimated accurately and it is not clear the proposed method performs better than RAPPOR in higher ranks. The major contribution of this study is an improvement of computation time at user and server side. It would be helpful for readers to show how much the computation time is improved compared to [4], particularly at the user side.

Reviewer 2



Private Heavy Hitters is the following problem: a set of n users each has an element from a universe of size d. The goal is to find the elements that occur (approximately) most frequently, but do so under a local differential privacy constraint. This is immensely useful subroutine, which can be used, e.g. to estimate the common home pages of users of a browswer, or words commonly used by users of a phone. In fact, this is the functionality implemented by the RAPPOR system in chrome, and more recently in use in iphones. To make things a bit more formal, there is an a desired privacy level that determines how well one can solve this problem. A \gamma-approximate heavy hitters algorithm outputs a set of elements that contains most elements with true relative frequency at least 2\gamma and a negligible number of elements with relative frequency less than \gamma. The best achievable accuracy is \sqrt{n log d}/eps, and Bassily and Smith showed matching lower and upper bounds. In that work, this was achieved at a server processing cost of n^{5/2}. This work gives two more practical algorithms that achieve the same gamma with a server processing cost of approximately n^{3/2}. On the positive side, this is an important problem and algorithms for this problem are already in largescale use. Thus advances in this direction are valuable. On the negative side, in most applications, the target gamma is a separate parameter (see more detailed comment below) and it would be much more accurate to treat it as such and present and compare bounds in terms of n, d and gamma. Clarifying these dependencies better would make the work more relevant and usable. Detailed comment: In practice, the goal is usually to collect several statistics at a small privacy cost. So a target eps, of say 1, is split between several different statstics, so that the target gamma is not \sqrt{n log d}/eps, but \sqrt{n log d}/ k\eps, when we want to collect k statistics. Often this k itself is determined based on what may be achievable, for a fixed constant gamma, of say 0.01. In short then, the gamma used in practice is almost never \sqrt{n log d}/eps, but a separate parameter. I believe therefore that the correct formulation of the problem should treat gamma as a separate parameter, instead of fixing it at ~\sqrt{n}.